# Refine, Don't Rewrite: The LearnFrom Framework for Consistency-Aware LLM Decompilation

## Abstract

Decompilation aims to translate binary executables into high-level source code; yet, the task remains demanding. Traditional tools, such as Ghidra, yield output that is structurally faithful but rarely recompilable or linkable. Recent methods built upon large language models improve readability and executability. However, unconstrained LLMs exhibit inherent stochasticity, they provide little guarantees of semantic consistency with the originating binary, which forces extensive cross-checking against the disassembly. We introduce the LearnFrom to address this challenge by leveraging traditional decompiler outputs and treating a code block corresponding to each control-flow graph node as a minimal editable unit to constrain modification scope. This patch generation design limits potential semantic deviations while reducing verification overhead. To achieve better model performance on the patch generation task in the context of decompilation, we further construct an open-source dataset of four million functions with explicit control-flow graph annotations, then use it to fine-tune the DeepSeek-Coder model series for specialized adaptation to the patch generation task. Within the Learn-From framework, highly preservation of CFG structural consistency is enforced throughout the editing process, ensuring reliable control-flow alignment. Under this constraint, identical base models achieve a 5% improvement in HumanEval re-execution rates over baseline systems, while the edit similarity is increased by 6%, with further gains of 8% when substituting DeepSeek-Coder V2 followed by re-fine-tuning. These results confirm that CFG-aligned constraints offer the structural reliability essential for large-scale reverse engineering, while still permitting further optimization as foundational models advance.

## 1 Introduction

Decompilation recovers source-level abstractions from binaries by rebuilding control flow, data flow, and types into a readable, high-level code. Decompilation supports a broad range of reverse-engineering tasks, from vulnerability discovery and malware analysis to program comprehension, making it an essential technique in the field.

Traditional decompilers such as Ghidra National Security Agency (2024) and RetDec Avast Software rely on control-flow structuring and data-flow recovery to emit pseudo-C that is primarily designed for human consumption. However, numerous studies Liu & Wang (2020); Wang et al. (2017) report that these outputs are rarely readily recompilable and typically require substantial manual repair before execution, limiting programmatic use. Motivated by these limitations, recent work employs large language models—via end-to-end generation, refinement of decompiler output, or recompilation-oriented feedback loops—to improve readability and, increasingly, re-executability.

Despite impressive performance metrics, current large language model (LLM) methods leave a practical concern largely unaddressed: semantic consistency between the recovered code and the originating binary, a gap exacerbated by the intrinsic randomness of LLM. Reverse engineering, particularly for algorithm reversal and security analysis, has zero tolerance for error. For instance, a seemingly intelligent optimization can corrupt the entire logic by failing to handle negative inputs correctly, such as a model replacing signed multiplication with a bitwise shift. In a security con-

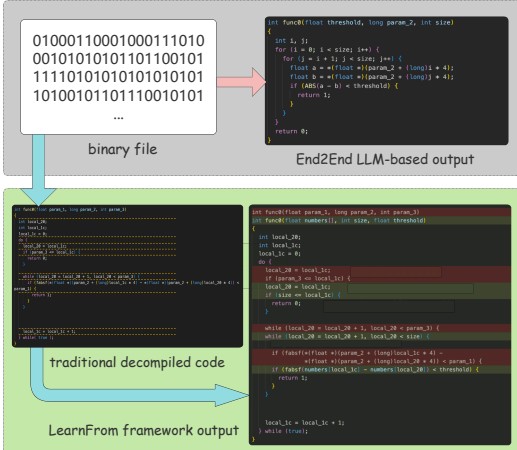

Figure 1: The upper panel illustrates the End2End LLM-based decompiler, the lower panel presents our LearnFrom approach. By integrating patch generation into the decompilation pipeline, the LearnFrom greatly reduces assembly-source verification effort.

text, an equivalent subtle error could neutralize a complex array bounds check, leading an analyst to approve vulnerable code and thereby invalidating the entire audit. Consequently, engineers must still audit the disassembly or decompiled code from traditional decompilers line-by-line, negating the promised efficiency gains.

A closer look at contemporary LLM-based pipelines helps explain the gap. Three dominant paradigms have emerged. First, exemplified by LLM4Decompile-End Tan et al. (2024), feeds assembly directly into a model that generates full C code in one pass. The second, adopted by DeGPT Hu et al. (2024) and the LLM4Decompile-Ref series, refines Ghidra's pseudo-C with an LLM, yielding marked gains in recompilability while retaining the static analyzer's structural precision. The third, represented by DecLLM Wong et al. (2025), loops between code generation and compilation or fuzzing; compiler diagnostics guide iterative fixes until all compiler tests pass. Although these diverse paths demonstrate an increasing awareness of the critical role of semantic consistency, their primary means of validation remain largely confined to techniques such as unit testing, fuzzing, and symbolic execution. While valuable for uncovering certain classes of errors, these methodologies are fundamentally limited by state explosion and are time-consuming, making it impractical to provide a sufficient security guarantee on their own. Therefore, rather than generating a flat sequence of executable code that requires extensive manual verification against low-level assembly, we aim to produce structurally reliable output that can be efficiently validated through limited comparison at the high-level language layer.

Therefore, we introduce the LearnFrom framework, which starts from the static output of a conventional decompiler and restricts the large language model to edit only the code basic block mapped to each control-flow graph (CFG) node, as shown in Fig.1. Each edit is strictly scoped to blocks, ensuring that the surrounding control structure and the original semantics remain intact. Experiments show that a domain-adapted model outperforms a generic counterpart on decompilation tasks, and that agent-based interactive iteration, as in approaches such as DeGPT, is likewise effective. We therefore combine both ideas, creating a 4-million-function code dataset in a fill-in-the-middle (FIM) task and fine-tune the DeepSeek-Coder series Guo et al. (2024) with LoRA Hu et al. (2022), endowing the resulting models with decompilation capabilities while preserving their natural-language conversational fluency.

Our approach ensures the integrity of the CFG structure while also improving the recompilation success rate, and because the fine-tuned model runs locally, it prevents proprietary code from leaving the premises.

This work makes three contributions.

- We present the LearnFrom framework, which couples CFG-indexed editing with a version control module and supports locally hosted models, turning one-off code regeneration into controlled, reversible code editing. LearnFrom preserves structural integrity and constrains semantic drift during iterative edits, greatly reducing verification effort.

- For achieving better model performance on the patch generation task in the context of decompilation, we built a four-million-function dataset enriched with CFG annotations and designed for a two-stage FIM task that first uses single-mask and then multi-mask objectives.

- We deliver a model that integrates decompilation and interactive optimisation in a single workflow, enabling practical, stepwise improvements while maintaining structural guarantees. When evaluated under these high semantic-consistency guarantees, our model delivers state-of-the-art recompilation performance.

## 2 RELATED WORK

Large language models Devlin et al. (2019); Shazeer et al. (2017), are now widely investigated as stand-ins or complements for traditional decompilers. LLM4Decompile offers an influential example: trained on paired C source and assembly, its End-to-End variant lifts binaries directly to C, whereas its Refined variant edits Ghidra output to improve identifier quality and high-level idioms. The two modes highlight a familiar tension: unconstrained generation yields fluent code but risks control-flow drift, whereas refinement-based methods, though starting from an existing skeleton, still provide no guarantee of structural consistency. Nova Jiang et al. (2023) advances direct lifting with hierarchical attention and contrastive objectives, improving Pass@1 and Pass@10 accuracy on several lifting tasks, though it still has deviations in loop and branch semantics that surface under symbolic execution.

To make decompiled programs runnable, subsequent work tightened the interface between static analysers and language models. DeGPT preserves Ghidra's skeleton but prompts a model to simplify types, merge redundant branches, and rename variables; human evaluators find the result markedly easier to read, although executability is not measured. DecLLM closes that gap by looping decompile edit compile until the patched file builds, feeding compiler diagnostics back into the model. Using GPT-4 Achiam et al. (2023), without any additional fine-tuning, DeGPT and DecLLM achieved performance close to the state of the art. However, repeated iterations on the code can lead to semantic drift, and the search cost grows with function size and can stall when error messages are misleading. Slade Armengol-Estapé et al. (2024) introduces intermediate-assisted decompilation: it first predicts a pseudo-high-level intermediate representation that captures control-flow edges, then realises the final C code in a constrained decoding step, reducing semantic drift without iterative search.

Insights from patch generation further inform decompilation. AlphaRepair Xia & Zhang (2022) tackles bugs through zero-shot infilling—given context, the model predicts the missing correct line—outperforming classic APR systems on Defects4J with no task-specific fine-tuning, though it relies on accurate localisation of the buggy region. ReCoder Zhu et al. (2021) generates syntax-guided edit scripts that operate on the abstract syntax tree (AST); every edit preserves AST well-formedness, so patches compile by construction and surpass statement-level baselines. RepairLLaMA Li et al. (2024) shows that minute LoRA adapters, trained on 4 k bug-fix pairs, can match far larger models, underscoring the power of task-specific adapters. SRepair Xiang et al. (2024) pushes efficiency by scoping edits to a single function: driven by failing tests, it fixes 300 of 522 Defects4J Just et al. (2014) bugs at an average cloud cost of three cents apiece and even repairs multi-function defects without explicit fault localisation.

Across these diverse systems, two recurring motifs emerge. First, constraining the model's action space tends to maintain syntactic and semantic validity – whether by generating structured edits or by scoping the task to a single function or line. This reduces the chance of the model drifting into irrelevant or invalid code. Second, using external feedback loops improves outcomes – be it compilation success signals, runtime tests, or other diagnostics that steer successive attempts. Techniques like DeGPT's error-informed loop and SRepair's use of test failures show that incorporating environment feedback helps the model converge to a correct solution instead of blindly trusting its first guess.

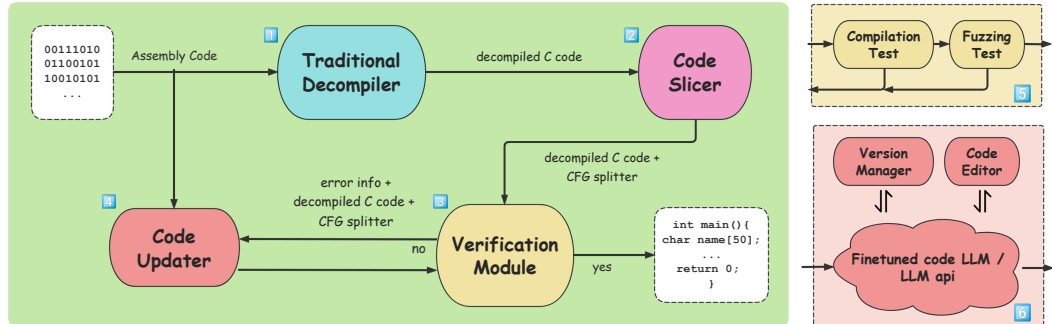

Figure 2: Overview of the LearnFrom framework.

# 3 METHOD

## 3.1 OVERVIEW

The overall workflow of LearnFrom is illustrated in Fig. 2. Given a binary executable, we first feed it to a conventional decompiler to obtain an initial pseudo-C. The code is passed to the Code Slicer module, which performs static analysis, reconstructs the control-flow graph, and inserts a comment token `cfg_splitter` at the first line of each code block corresponding to a CFG node—hereafter referred to as a basic block —while simultaneously assigning every basic block a unique identifier. Practical considerations are detailed in CFG Extraction section, such as how to treat comments, blank lines and other non-semantic elements. The annotated code is then subjected to ordinary compilation test and to libFuzzer fuzzing test; if both checks succeed the pipeline terminates, otherwise it proceeds to the next stage. At this point the LearnFrom model receives a compound prompt that includes the cfg-splitter-tagged decompiler output, the original assembly for context and the current compilation error messages. The model is first asked to identify the blocks that require modification and then, through an external script (Code Editor), to rewrite only those blocks. Whenever the LLM produces a new basic block, we rigorously verify that the occurrence and placement of branch keywords—such as `if` and `while`—match corresponding original block. The revised programme is compiled and fuzzed once more; should it still fail, control passes to Version Manager that enforces code version control designed to prevent the model from drifting through repeated erroneous edits and to limit resource consumption. Once the code passes both compilation tests and fuzzing tests, it is emitted as the final output; during testing, the code is fed into the test-case module.

## 3.2 CODE SLICER

Currently, no off-the-shelf tool can comprehensively map CFG nodes to their corresponding code blocks in C and C++. We therefore developed Code Slicer, which reports the first-line number and an index for each block, allowing LLMs to locate code problems quickly, telling LLMs the minimal editable unit. It parses the listing with an incremental syntax engine rooted in the Tree-Sitter grammar for C and C++, walks the resulting concrete syntax tree, and recognises every construct that delimits control flow—conditional branches, loops, switch cases, early returns, labelled statements, and nested compound blocks. For each such boundary it records the first source line belonging to the block and inserts a lightweight comment token that later serves as an anchor.

## 3.3 FINE-TUNING

An agent composed of Code Slicer, Version Manager and a lightweight external editing script, paired with a general LLM, can reproduce the full LearnFrom workflow. However, most general-purpose and code-oriented large LLM lack native decompilation capabilities, and their re-execution success rate within LearnFrom still lags behind that of highly specialised systems such as LLM4Decompile. Paradoxically, LLM4Decompile's extensive task-specific training leaves it unable to interpret compiler diagnostics, so it cannot drive the iterative repair loop required by LearnFrom.

We assembled a parallel corpus of source code and its corresponding assembly output and then fine-tuned several code LLMs with a parameter-efficient LoRA scheme, which lowers training overhead while preserving the model's capacity for interactive refinement. Each source-assembly pair is reformulated as a fill-in-the-middle (FIM) reconstruction task: Code Slicer annotates every control-flow graph (CFG) node with a cfg-splitter tag; a subset of those blocks is masked, while the compiled assembly is supplied as additional context. The model must therefore infer and restore the missing fragments with instruction-level fidelity.

To sharpen the model's grasp of program structure, we retain the CFG annotations during fine-tuning. The annotation allows the LLM to reason over the relationships between execution paths and their semantic implications rather than merely imitating surface syntax, helping LLM to outperform in this specific task. The curriculum progresses in two stages of increasing difficulty. During the initial phase the model sees single-mask samples, mirroring the pre-training paradigm of DeepSeek-Coder and ensuring a smooth hand-over to our task domain. The second phase raises the bar by masking multiple, interdependent blocks within the same function, thereby forcing the network to co-ordinate several local reconstructions under a global consistency constraint and significantly strengthening its reasoning capability.

When deciding how to expose the assembly context, we compared two alternatives: (i) indexing the assembly into an external, queryable memory and (ii) injecting the raw assembly directly into the prompt. Because typical functions produce a manageable amount of machine code—and the second option reduces system complexity—we adopt the direct-injection strategy. Empirically, this assembly-aware reconstruction mechanism enables the model to recover developer intent more faithfully than naive text-only variants.

## 4  EXPERIMENTS

### 4.1  CFG EXTRACTION

Our pipeline begins with a structural analysis pass that converts each compilation unit into a control-flow graph view whose vertices correspond exactly to source-level basic blocks. To recover the blocks, we parse the original `.c` or `.cpp` file with the *tree-sitter* incremental parsing library, whose C API and Python bindings expose a full concrete syntax tree with constant-time random access to node boundaries. We configure the parser with the official `tree-sitter-cpp` grammar. The parse tree is then traversed in preorder; whenever the visitor encounters a node whose `type` matches an entry in the hand-curated set, including `if_statement`, `while_statement`, `for_statement`, `switch_statement`, and other control-flow statements. The opening line number of that node is recorded as a split point. Edge cases are handled explicitly to ensure that every recorded split line corresponds to a semantically meaningful basic block. First, non-semantic tokens—braces, null statements, comments, and all pre-processor directives—are skipped so that purely syntactic elements never define a block boundary. Second, consecutive declaration or expression statement nodes are merged into a single fall-through block, preventing an explosion of one-line blocks in straight-line code. Third, labeled statements are treated specially: the label itself is ignored, but the statement that follows is promoted to its own block so that indirect jumps remain well-formed in the reconstructed CFG. Pragma-style annotations such as `#type` are currently ignored, and no block tag is emitted for them. Nested compound statements are traversed recursively, and any duplicate line numbers collected from overlapping walks are removed at the end, guaranteeing a strictly increasing, unique sequence of split points. Finally, if the parser encounters malformed input, the analyser returns an empty split list together with an error flag, allowing downstream components to fall back to the original decompiler output without interrupting the pipeline.

### 4.2  DATASETS

The corpus is built from decompile bench, a parallel collection of source files and their compiled assembly, stored as arrow records that contain the source string, the original file path, an optional symbol name and the corresponding assembly. Preprocessing proceeds in a uniform pipeline for C and C++, with a minor difference in the front-end used to recover control-flow boundaries. Each record is first normalised by extracting the `code`, file-suffix and `asm` fields. The suffix determines

```
input:
Context: A C/C++ source code with
several lines obscured, control-
flow split points are tagged with
"//<cfg_splitter>".   The
complete assembly generated from
the unobscured version of that
source. Task: Use the assembly
and the CFG split tags to infer
the original C/C++ code for each
masked block. Output the
recovered blocks in numerical
order.

MASKED_CODE:
{masked code}

ASSEMBLY:
{assembly code}
```

```
output:
{recovered code}
```

```
{assembly code}
pushq %rbp
movq %rsp, %rbp
subq $0x570, %rsp              # imm = 0x570
movq %rdi, -0x8(%rbp)
movq %rsi, -0x10(%rbp)
movq -0x8(%rbp), %rax
movq %rax, -0x548(%rbp)
leaq -0x1b0(%rbp), %rdi
movq %rdi, -0x540(%rbp)
callq 0xf97c0
movq -0x548(%rbp), %rdi
movq -0x540(%rbp), %rsi
callq 0x6703a0
...
```

```
{recovered code}
<|fim_holestart|>    ExceptionOr<DepT> depResult;
    getDepResult(depResult);<|fim_holeend|>
```

```
{masked code}
<|fim_begin|>void getImpl(ExceptionOrValue& output) override {
    typedef _::FixVoid<_::ReturnType<Func, _DepT>> T;
    typedef _::FixVoid<_DepT> DepT;
<|fim_hole|>

    KJ_IF_SOME(depException, depResult.exception) {//<cfg splitter>
      output.as<T>() = ExceptionOr<T>(false, kj::mv(depException));//<cfg splitter>
    } else KJ_IF_SOME(depValue, depResult.value) {//<cfg splitter>
      output.as<T>() = handle(MaybeVoidCaller<DepT, T>::apply(func, kj::mv(depValue)));//<cfg splitter>
    }
  }<|fim_end|>
```

Figure 3: It presents an example from the training set, illustrating the dataset's composition.

which parser is applied, while headers or unsupported suffixes are skipped. The extractor returns an ordered list of source line numbers, each marking the first line of a block.

We traverse the block identifiers to obtain the index and line range of every region, yielding a deterministic mapping from each identifier to a half-open interval that defines what can be masked. Masking policies differ only in the number of blocks removed: the single-mask phase randomly drops one block, whereas the multi-mask phase omits 15 % of the blocks, rounding up when fewer than one would otherwise be selected. If no split points are detected, the entire file is replaced by a single mask. When constructing the masked source, the covered lines are deleted and a placeholder is left on each block's first line; blank lines from the original file are preserved. Because real-world code carries irregular line endings and erratic indentation, additional normalisation steps are applied to ensure stable parsing. Each data instance,illustrated in Fig.3, is divided into an input part and an output part. The input part begins with a brief natural-language preamble that states the task, followed by two components: the masked source fragment delimited by FIM markers and a verbatim copy of the assembly emitted from the unmasked code. The output part lists the masked basic blocks in their original order. All placeholders are standardised to a single hole token to reduce lexical fragmentation, and the ground-truth blocks are concatenated in ascending block-index order so their original sequence is implicit. Both training stages are produced by the same pipeline. The first stage, limited to single masks, stabilises learning and echoes the model's pre-training regime, offering short, well-anchored snippets whose recovery depends mainly on local context and explicit control-flow cues. The second stage increases difficulty by masking multiple, often non-contiguous blocks in the same function, forcing the network to use CFG anchors and assembly guidance for globally consistent reconstruction. For both phases the corpus is split at file level in a 15 : 1 : 1 ratio into the respective training sets and a shared validation set, with ExeBench Armengol-Estapé et al. (2022) and HumanEval-Decompile held out for testing. Strict checks ensure that any newly generated block preserves the presence and placement of branching keywords such as if and while, guaranteeing high CFG-level consistency.

HumanEval-Decompile is a compact benchmark distilled from the 164 tasks of OpenAI's HumanEval suite: each Python reference solution was mechanically translated into a single, self-contained C function together with its original assertions. Every task depends only on the standard C library, so the code compiles out-of-the-box with GCC and isolates pure algorithmic reasoning without project-specific baggage. ExeBench, by contrast, draws 5,000 real-world C functions mined from diverse GitHub projects and paired with I/O examples. The corpus retains user-defined structures, type aliases and helper routines, offering a much richer syntactic and semantic landscape than HumanEval-Decompile and mirroring production code complexity across nine established metrics.

These characteristics make ExeBench a tougher test of decompilation: successful recovery requires not only control-flow fidelity but also accurate reconstitution of custom data types and external interactions.

### 4.3 CONFIGURATION

We fine-tune the models in two curriculum stages that mirror the construction of the corpus. In both stages the backbone weights remain frozen and only LoRA adapters are updated. Unless otherwise stated, all runs use bf16 arithmetic, AdamW with a fixed learning rate of $1 \times 10^{-4}$, gradient clipping at 1.0, a warm-up ratio of 0.1, per-device batch size of 8 sequences, gradient accumulation of 8 steps, and a maximum sequence length of 1,024 tokens. We employ gradient checkpointing and DeepSpeed ZeRO-2 sharding to fit the memory footprint while keeping optimiser states offloaded to host memory when beneficial. The LoRA configuration follows the script: rank r = 8, $\alpha = 32$, dropout 0.1.

Experiments are executed on two GPU pools: a node equipped with eight NVIDIA A100 40,GB cards and a second node with four NVIDIA A800 80,GB cards. The A100 node is used for most ablations and all StageI runs for both backbones. The A800 node provides the extra headroom needed for the largest configuration on DeepSeek-Coder-v2 Zhu et al. (2024); with the above hyper-parameters, this job requires nine days of wall-clock time from the start of StageI to the end of Stage II, including validation and checkpointing. Smaller variants, such as the v1 backbone or reduced curriculum lengths, complete proportionally faster on the A100 pool. To avoid I/O contention on the shared filesystem we redirect all HuggingFace, Torch and CUDA caches to a dedicated NVMe path.

Convergence is tracked on the held-out validation split from each stage. StageI reduces validation perplexity rapidly over the first epoch and then saturates, at which point we transition to StageII using the best StageI adapter as initialisation. StageII exhibits slower but steady improvements, consistent with the increased difficulty of multi-mask reconstruction. The curriculum consistently yields the best recompilation success when both stages are completed; omitting StageI or training StageII from scratch degrades performance.

| Model/Benchmark | HumanEval-Decompile | | | | | ExeBench | | | | |
|---|---|---|---|---|---|---|---|---|---|---|
| | O0 | O1 | O2 | O3 | AVG | O0 | O1 | O2 | O3 | AVG |
| LLM4Decompile-End-6.7B | 68.05 | 39.51 | 36.71 | 37.20 | 45.37 | 22.89 | 16.60 | 16.18 | 16.25 | 17.98 |
| Ghidra+LLM4Decompile-Ref-1.3B | 68.90 | 37.20 | 40.85 | 37.20 | 46.04 | - | - | - | - | - |
| Ghidra+LLM4Decompile-Ref-6.7B | 74.39 | 46.95 | 47.56 | 42.07 | 52.74 | - | - | - | - | - |
| DecLLM | 80.49 | 65.24 | 65.5 | 58.5 | 67.43 | - | - | - | - | - |
| Ghidra+LFMv1-1.3B+LFW | 65.54 | 45.90 | 46.61 | 37.70 | 48.94 | 24.44 | 17.97 | 17.83 | 16.60 | 19.21 |
| Ghidra+LFMv1-6.7B+LFW | 87.39 | 49.53 | 48.16 | 44.88 | 57.49 | 2.87 | 22.53 | 22.24 | 22.10 | 24.19 |
| Ghidra+LFMv2+LFW | **88.60** | 55.57 | 52.45 | **48.06** | **61.17** | 33.55 | **22.72** | 22.31 | 22.26 | 25.21 |
| IDA Pro+LFMv2+LFW | 88.21 | **58.76** | **53.80** | 42.77 | 60.89 | **34.10** | 22.20 | **22.97** | **22.99** | **25.57** |
| IDA Pro+Claude 4.0+LFW | 50.57 | 48.04 | 49.05 | 33.18 | 45.21 | 20.09 | 16.19 | 15.48 | 15.66 | 16.86 |

Table 1: Comparison of each method's re-execution rate across different test sets and compiler-optimization levels. The + sign indicates that the preceding decompiler is used together with the LearnFrom framework. LFM stands for LearnFrom Model, i.e., the model we obtained by fine-tuning with our method, whereas v1 and v2 refer to different versions of the DeepSeek Coder base model.

### 4.4 RESULTS

While LearnFrom's chief contribution is to guarantee trustworthy CFG alignment in LLM-based decompilation, we can still compare its effectiveness with prior work by evaluating the re-execution rate—the proportion of recovered functions that both compile and pass the original I/O test harness—on HumanEval-Decompile and ExeBench across optimisation levels O0 to O3. Because the framework enforces strict basic-block boundaries, the recovered code already preserves CFG fidelity; in our view, meeting this level of semantic consistency is a prerequisite rather than a performance target, so traditional edit-similarity metrics, which is used in earlier work, are no longer informative and are therefore omitted.

| Model/Optimization | O0 | O1 | O2 | O3 | AVG |
|---|---|---|---|---|---|
| Ghidra+LLM4-Decompile-Ref-6.7B | 15.59 | 13.53 | 13.42 | 12.73 | 13.82 |
| CIM 6.7B+CIW | 49.16 | 40.25 | 39.54 | 39.02 | 41.99 |
| Ghidra+LFMv1+LFW | 48.77 | 42.53 | 36.96 | 37.15 | 41.35 |

Table 2: Main comparison of Edit Similarity with different methods on the HumanEval-Decompile benchmark, where L2R denotes a fine-tuning task implemented in the conventional left-to-right next-token prediction format.

| Model/Optimization | O0 | O1 | O2 | O3 | AVG |
|---|---|---|---|---|---|
| Ghidra+LLM4-Decompile-Ref-6.7B | 74.39 | 46.95 | 47.56 | 42.07 | 52.74 |
| Ghidra+DeepSeek-Coder-6.7B+LFW | 58.52 | 33.97 | 31.56 | 31.95 | 39.00 |
| Ghidra+DeepSeek-Coder-6.7B-L2R+LFW | 76.02 | 47.50 | 47.97 | 42.43 | 53.48 |
| Ghidra+LFMv1+LFW | 89.39 | 47.53 | 48.16 | 44.88 | 57.49 |

Table 3: Main comparison for the ablation study, where L2R denotes a fine-tuning task implemented in the conventional left-to-right next-token prediction format.

We first reproduce LLM4Decompile-End, the current open-source state of the art for End-to-End binary decompilation. On HumanEval-Decompile, 6.7 B model attains an average re-execution rate of 45.37 %, and only 17.98 % on ExeBench. Adding Ghidra's structured output as a front-end raises the average to 52.74 %, confirming earlier observations that injecting coarse syntactic hints helps large sequence models align with compiler idioms .

We fine-tune DeepSeek-Coder checkpoints with Low-Rank-Adaptation (LoRA) layers using our four-million-function, CFG-annotated FIM corpus. With LoRA, the 6.7 B model already surpasses the reference by +4.9 % absolute on HumanEval, confirming that parameter-efficient tuning can endow a code LLM with decompilation priors at modest cost . Our 6.7 B-parameter system surpasses the strongest available 22 B end-to-end model in re-execution performance. A lightweight 1.3 B variant remains competitive, trailing the 6.7 B LLM4Decompile reference by just 3.8 %.

To maximise performance, we fine-tuned the 16 B parameter DeepSeek-Coder-V2-Lite-Instruct model, paired it with the state-of-the-art IDA Pro decompiler, and integrated it into the LearnFrom framework, securing an improvement of nearly 10 % while maintaining high CFG structural consistency. On HumanEval-Decompile the fine-tuned DeepSeek-Coder-V2 with LearnFrom reaches 61.17 % average re-execution, a +15.8 pp improvement over the best end-to-end model. ExeBench rises to 25.57 %. When we replace our tuned model with Claude 4 Opus and run it through Learn-From, performance climbs from 21.22 % to 45.21 % on HumanEval, closely matching the 46.04 % achieved by the much larger 1.3 B LLM4Decompile-Ref. This confirms that LearnFrom's block-scoped repair loop is itself responsible for roughly half of the observed accuracy gains.

We also conducted experiments on code edit similarity. As shown in Tab. 2, which presents the edit similarity between decompiled and source code, our method significantly outperforms the baseline. It is worth noting that CIM and CIW Liu et al. (2025), a paper that has not undergone peer review, report a slightly higher similarity score than ours. However, their work does not address CFG-level consistency.

## 4.5 ABLATION STUDY

We conducted a series of ablations to isolate the contribution of individual components. First, to evaluate the efficacy of our fill-in-the-middle formulation we replicated the data construction strategy of LLM4Decompile but substituted LoRA fine-tuning so that the base model retained interactive editing capability . Under identical hyper-parameters, the LoRA variant improves re-execution by two percentage points over the original LLM4Decompile reference, whereas our block-aligned FIM

| Model/Benchmark | O0 | O1 | O2 | O3 | AVG |
|---|---|---|---|---|---|
| LFMv1 w/o Fuzzing | 83.83 | 48.77 | 43.98 | 42.52 | 54.78 |
| LFMv1 w Fuzzing | 87.39 | 49.53 | 48.16 | 44.88 | 57.49 |

Table 4: Ablation study on fuzzing test module.

task yields the larger gains reported in Tab. 3, confirming that structural cues—not merely additional training—drive the improvement.

We removed the fuzz-testing stage from the validation loop and relied exclusively on compilation feedback. Many patches that passed the compiler subsequently crashed when re-executed, lowering the overall re-execution rate by three percentage points, as in Tab. 4, and confirming prior work that fuzzing is indispensable for revealing latent semantic defects that elude static checks .

We also disabled the Version Manager, allowing the agent to iterate without checkpoint control. Re-execution variance increased sharply and wall-time nearly doubled, indicating that disciplined state management is critical for stable convergence, in line with recent observations on iterative code-repair pipelines . Finally, we replaced the external code editor with a prompt that asked the language model to "use the fewest edits possible." Although this shortcut occasionally sped up convergence, the model still altered CFG structure or semantics in a substantial fraction of cases, even when explicitly instructed to avoid such changes, echoing earlier findings on the limits of prompt-only control .

Taken together, these ablations show that the gains stem from the joint effect of CFG-constrained masking, fuzz-based validation, and version-controlled iteration. Semantic fidelity to the original program is treated as a prerequisite rather than a tunable metric; therefore we do not report edit-similarity scores that earlier studies used for looser, token-level comparisons.

## 5 CONCLUSION

In summary, LearnFrom demonstrates that constraining a fine-tuned code model with CFG granularity, staged FIM curricula, and a limit–rollback schedule can systematically raise recompilation success—delivering 5 pp and 8 pp absolute gains on HumanEval Chen et al. (2021) when applied to DeepSeek-Coder v1 and v2 respectively, while retaining the readable output that recent LLM-based decompilers have highlighted as their chief advantage. At the same time, LoRA adaptation keeps training economical and privacy-preserving on commodity clusters, confirming previous observations about its efficiency in large-model fine-tuning.

Although these results close much of the gap between classical tools—whose strength lies in structural fidelity—and end-to-end neural synthesis, they do not yet match the interactive conveniences reverse engineers rely on daily: mainstream decompilers such as Ghidra and Hex-Rays let analysts jump from a high-level variable to its exact machine-level definition, preserving a one-to-one mapping between symbolic names and assembly addresses for rapid cross-reference. Enabling the same clickable correspondence in a learned editing pipeline remains an open challenge; doing so will likely require augmenting the language model with a symbolic store of variable lifetimes and pointer provenance or with a post-processing layer that reconciles generated identifiers against the original program's data-flow. We leave this integration of semantic links—and its attendant UI affordances—as future work, anticipating that a tight loop between learned refinement and source–binary cross-navigation will be the next decisive step toward a practical, engineer-grade neural decompiler.

## 6 STATEMENT

### 6.1 ETHICS STATEMENT

This work adheres to the ICLR Code of Ethics. The research is entirely computational and does not involve human subjects, sensitive personal information, or the collection of private data. The

datasets used are publicly available and have been widely employed in the research community, and we followed responsible practices in data usage and documentation. The methods proposed are intended for scientific research and education, and we are not aware of any immediate harmful applications. We have made efforts to avoid introducing or amplifying unfair bias and to ensure that the results respect principles of fairness, integrity, and transparency. The authors declare that there are no conflicts of interest or external sponsorships that could compromise the integrity of this work.

## 6.2 REPRODUCIBILITY STATEMENT

We have taken active measures to ensure the reproducibility of our results. All key details of the proposed method, including model architectures, training procedures, hyperparameters, and evaluation metrics, are described in the main paper and appendix. To further facilitate reproducibility, we will provide an anonymous link to the source code and scripts used for data preprocessing, model training, and evaluation in the supplementary material. We include a clear description of data processing steps for all datasets used in experiments. Together, these resources should enable independent researchers to reproduce and verify our findings.

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

# A  APPENDIX

## A.1  LLM USAGE STATEMENT

Large language models were employed in this work only in a limited and carefully managed capacity. Their use was confined to linguistic refinement, including grammar checking, stylistic polishing, and the improvement of textual fluency. In addition, LLMs were cautiously utilized to generate a few minor auxiliary code modules and simple test scripts, but they played no role in shaping the research questions, designing or implementing the central methods, executing experiments, or analyzing results. The intellectual conception of the study, the methodological framework, the core system development, and the interpretation of findings were conducted solely by the authors. All content produced with the assistance of LLMs was rigorously reviewed and revised to ensure both accuracy and originality. The authors accept full responsibility for the manuscript in its entirety, and, consistent with ICLR policy, LLMs are not recognized as contributors or authors of this work.

## A.2  VERSION MANAGER

During iterative repair we consider two principal revision paradigms. The first relies exclusively on compiler diagnostics: the model emits large numbers of candidate patches, each is compiled and, when possible. The second paradigm spawns a branch for every edit—much like a Git workflow—and delegating to a supervisory agent the choice of which branch to pursue or abandon. Our preliminary experiments showed that LLM appears unable to decide which branch is closest to the correct solution, choosing branches randomly; they may even propagate errors by iterating on incorrect branches, and excessively long contexts lead to performance degradation. It touches an ever larger set of blocks and consumes disproportionate time. And because our setting constrains code changes to pre-defined basic blocks, the search space is modest, the former mode's time cost is acceptable.

Therefore, we integrate the two modes and design a Version Manager: the LLM can decide whether it would edit on the last version or rollback to the pristine baseline, and it is allowed to stay on its editing branch for a maximum of three consecutive edits, and if compilation still fails, the system rolls back to the pristine baseline; overall, no more than four such rollbacks are permitted during evaluation. In practice this cap yields a favourable balance between search depth and reliability, delivering the desired accuracy without excessive resource expenditure.

