# OpenReview forum: "Refine, Don’t Rewrite: LearnFrom for Consistency-Aware LLM Decompilation"
_ICLR.cc/2026/Conference — ICLR 2026 Conference Withdrawn Submission_

### Official Review · Reviewer_3GE2 · 2025-10-25

**Soundness:** 3
**Presentation:** 2
**Contribution:** 3
**Rating:** 4
**Confidence:** 4

**Summary:**

This paper works on binary decompilation. Traditional decompilers like Ghidra preserve structure but seldom produce recompilable output, while recent large language model (LLM) approaches improve readability yet lack semantic consistency with the original binary. The authors propose LearnFrom, a framework that constrains LLM-based code editing using control-flow graph (CFG) nodes as minimal edit units, ensuring structural and semantic alignment. They also introduce a four-million-function dataset with explicit CFG annotations to fine-tune the DeepSeek-Coder models for this task. Experiments show that LearnFrom improves HumanEval re-execution rates by 5% and edit similarity by 6%, with further gains of 8% using DeepSeek-Coder V2 after re-finetuning. These results demonstrate that CFG-aligned constraints enhance the reliability of LLM-driven decompilation, offering a path toward more consistent and verifiable reverse engineering.

**Strengths:**

The paper introduces a framework for performing fragment-level edits on decompiled code. This approach enables localized modifications within specific code regions, rather than re-editing the entire decompiled output, which helps minimize unintended changes and preserve structural consistency.

**Weaknesses:**

There're few points need to be clarify before making a final decision.

## Clarity and Methodological Detail
The paper’s methodology is described in only one page, leaving many critical implementation details unclear. Key questions remain unanswered:

How exactly are the compilation test and libFuzzer-based fuzzing (mentioned on Page 4, Line 187) integrated into the workflow?
What is the setup process for the compilation and fuzzing environments, especially for complex real-world programs with diverse dependencies?

## Role of Validation Constraints in Code Evolution
It is unclear how the two validation constraints (compilation success and fuzzing correctness) guide the iterative refinement process. Are they used merely as binary (pass/fail) signals, or do error messages from the compiler or fuzzer actively inform and steer the code evolution? If the latter, what mechanisms translate these diagnostics into concrete edits?

## Error Localization and Repair Strategy
The paper states that code revision operates block-wise, guided by control-flow graph (CFG) node locations extracted via Tree-sitter. However, this raises several concerns:

What happens if the decompiled code cannot be parsed by Tree-sitter (e.g., due to syntax errors or unconventional constructs)?
Since both validation steps require successful compilation, how does the system handle cases where the initial decompiled output is uncompilable?
In decompiled code riddled with low-level control-flow artifacts (e.g., goto statements, infinite loops like while(1)), can block-wise editing meaningfully reconstruct high-level, readable logic?

## Mismatch Between Training and Evaluation Data
The model is trained on pairs of assembly and original source code, yet the refinement and evaluation are performed on pseudo-code generated by decompilers. This introduces a domain gap: how does a model trained on clean source code generalize to the noisy, often syntactically irregular output of decompilers? The paper does not justify this design choice or analyze its impact.

**Questions:**

Please refer to weakness.

Where are the code and the models?

---

### Official Review · Reviewer_Xcer · 2025-10-27

**Soundness:** 2
**Presentation:** 2
**Contribution:** 2
**Rating:** 2
**Confidence:** 4

**Summary:**

a)	This paper proposes LearnFrom: a "Refine, Don't Rewrite" LLM decompilation framework. The core idea is to use CFG basic blocks as the smallest editable unit, performing constrained patch-style editing of the pseudo-C code generated by traditional decompilers (such as Ghidra/IDA), thereby reducing verification costs while suppressing semantic drift. To improve patch generation capabilities, the authors constructed a four-million-function, CFG-annotated FIM (fill-in-the-middle) corpus and fine-tuned the DeepSeek-Coder family with LoRA to obtain a domain adaptation model (LFM). Under strong CFG consistency constraints, the authors report improvements in both HumanEval-Decompile re-execution rate and edit similarity, with further gains achieved when re-tuning on a v2 base.

**Strengths:**

- The paper proposes a controlled editing approach using structural constraints and version rollbacks effectively suppresses semantic drift and reduces verification costs.
- The paper construct the training data using CFG and FIM, which is highly targeted.

**Weaknesses:**

- The experimental results lack statistical and uncertainty information, and the analytical details presented are insufficient. Furthermore, the source details, licenses, deduplication, and isolation strategies for the four million functions from the test set need to be more transparent. The dataset may also require more detailed disclosure. While training computing power and duration are reported, the average number of iterations, rollbacks, and fuzz budget during the inference phase are not quantified.
- The paper does not propose new learning objectives, structured decoders, formal semantic constraints, or verifiable reasoning mechanisms. The so-called "semantic consistency control" is essentially implemented by limiting the edit scope and checking branch keyword positions using CFG basic blocks, which is structural anchoring rather than semantic equivalence.
- The experimental design lacks a strong semantic alignment metric (only compilation and I/O harness pass), making it difficult to support the claim that the modification can reliably preserve semantics. Furthermore, the lack of statistical robustness limits the credibility of the conclusions.
- The base model (DeepSeekCoder-V2) is relatively out-of-date. More recent open models like Qwen3.
- The experiments lack more recent baseline models.
- Please mind the difference between \citet and \citep.

**Questions:**

- Can you provide 95% CIs and paired tests for the main comparisons in Tables 1 and 3? This would significantly enhance the confidence of the conclusions.
- In addition to keyword alignment, have you performed symbolic execution/path equivalence verification on a sampled dataset? If so, please provide the distribution of failure types.
- How do you ensure the source/licensing and deduplication strategy for the millions of functions are consistent with HumanEval-Decompile and ExeBench? Can you provide more details?
- What are the average number of iterations/rollbacks and fuzzing time on the earnFrom inference side? Are there any results on the sensitivity of hyperparameters to convergence speed?

---

### Official Review · Reviewer_VCtt · 2025-10-31

**Soundness:** 3
**Presentation:** 3
**Contribution:** 2
**Rating:** 4
**Confidence:** 2

**Summary:**

LearnFrom proposes a CFG-constrained “patch generation” framework for LLM-based decompilation. Instead of end-to-end code synthesis, it edits only basic blocks aligned to CFG nodes in decompiler output, with hard checks that branch keywords and block placements remain consistent; compilation and libFuzzer tests guide iterative fixes under a rollback-capped “Version Manager.” They also build a ~4M-function CFG-annotated FIM dataset and LoRA-tune DeepSeek-Coder variants. On HumanEval-Decompile and ExeBench, they report higher “re-execution” (compile+tests pass) rates than several baselines, especially when pairing IDA/Ghidra with their fine-tuned model and fuzzing.

**Strengths:**

On HumanEval-Decompile, LFMv2+LFW averages 61.17% vs 52.74% for Ghidra+LLM4Decompile-Ref-6.7B; ExeBench rises to ≈25.57%.

**Weaknesses:**

Several comparisons mix different decompilers (Ghidra vs. IDA) and models (Claude, DeepSeek, LLM4Decompile) with/without LearnFrom. It’s hard to isolate the contribution of CFG-scoped editing vs. stronger front-ends; an apples-to-apples grid (same decompiler, same backbone, ±LearnFrom) is only partially shown.

The best setting yields ≈25% re-execution, which suggests limited practicality for complex, real-world C—this deserves more analysis (failure modes, types/structs handling).

**Questions:**

Your branch-keyword check ensures if/while presence/placement, but how do you guard semantic equivalence for short-circuiting, switch fall-through, and early returns? Any cases where syntax passes your check yet CFG still changes?

For large functions, when the raw assembly plus masked code exceed context, what truncation or windowing strategy is used? Any measured failure rate vs. function size?

Could you add rows with the same decompiler and same base model ±LearnFrom (and ±fuzzing) to quantify the isolated contribution, especially on ExeBench?

---

### Official Review · Reviewer_9Cji · 2025-10-31

**Soundness:** 2
**Presentation:** 1
**Contribution:** 3
**Rating:** 4
**Confidence:** 3

**Summary:**

Rule-based decompilation tools (e.g. Ghidra, RetDec) are accurate in structure.
Recent LLM-based decompilers generate cleaner code but can change the meaning (semantic drift). This work proposes LearnFrom. This approach:
- starts with a traditional decompiler output.
- Lets an LLM edit basic blocks instead of rewriting everything.
- Uses extensively tooling to improve the verification of each edit (comilation tests, fuzzing, etc).

The authors fine-tune DeepSeek-Coder with LoRA on 4 million functions annotated with CFG data, using FIM.

Ultimately, this improves decompilation results.

**Strengths:**

- Comprehensive ablation study, this is much appreciated.
- The code slicer, version control, fuzzying, etc modules are very interesting and useful on their own.
- Clever, effective use of FIM and LoRA.
- Comprehensive evaluation.

**Weaknesses:**

- The evaluation can be misleading: re-execution rate does not imply correctness. Any insights or analysis on whether this metric is working well here? "Semantic fidelity to the original
program is treated as a prerequisite rather than a tunable metric; therefore we do not report editsimilarity scores that earlier studies used for looser, token-level comparisons." The emphasis of this work on semantic correctness is appreciated, but isn't this a very strong statement given that you have no formal correctness guarantees? How good are the unit tests?
- Presentation and writing: Figure 1 is important but impossible to read. Figure 2 is also hard to read should have a caption explaining the figure. The figures in general are hard to read. In the abstract, there are some sentences that are hard to read.
- Inaccurate representation of related work: For example, Slade "introduces intermediate-assisted decompilation: it first predicts a pseudo-high-level intermediate representation that captures control-flow edges, then realises the final C code in a constrained decoding step, reducing semantic drift without iterative search". This is not at all what Slade does. A more accurate description would be the one you used for LLM4Decompile:  "feeds assembly directly into a model that generates full C code in one pass". In fact, Slade predates LLM4Decompile. Earlier in the introduction, when you say " Three dominant paradigms have emerged. First, exemplified by LLM4Decompile-End Tan et al. (2024), feeds assembly directly into a model that generates full C code in one pass". Here, and BTC [1] and Slade could be cited as earlier examples of this paradigm (in fact, LLM4Decompile cites them).




[1] https://arxiv.org/abs/2212.08950



Other details:
- The citations don't have the right format.
-  “highly preservation"
- Occasionally misses articles.

**Questions:**

- What's the exact input the system receives? What assembly format?
- The code slicer is interesting, can you give more details about its implementation?
- "When deciding how to expose the assembly context, we compared two alternatives: (i) indexing
the assembly into an external, queryable memory and (ii) injecting the raw assembly directly into
the prompt. Because typical functions produce a manageable amount of machine code—and the
second option reduces system complexity—we adopt the direct-injection strategy." how would scale this to long basic blocks? E.g., unrolled loops.
- How does CFG-scoped editing handle obfuscated or optimized binaries where basic blocks don’t map neatly?
- Will this be open-sourced in any way?
- How does LearnFrom compare to simpler prompt-only methods (without fine-tuning)?

---

### Note · Authors · 2025-11-26

I have read and agree with the venue's withdrawal policy on behalf of myself and my co-authors.